# Functionalized Upconversion Nanoparticles for Targeted Labelling of Bladder Cancer Cells

**DOI:** 10.3390/biom9120820

**Published:** 2019-12-03

**Authors:** Dmitry Polikarpov, Liuen Liang, Andrew Care, Anwar Sunna, Douglas Campbell, Bradley Walsh, Irina Balalaeva, Andrei Zvyagin, David Gillatt, Evgenii Guryev

**Affiliations:** 1Faculty of Medicine and Health Sciences, Macquarie University, Sydney, NSW 2109, Australia; dmitry.polikarpov@hdr.mq.edu.au (D.P.); liuen.liang@mq.edu.au (L.L.); david.gillatt@mq.edu.au (D.G.); 2Centre for Nanoscale Biophotonics, Macquarie University, Sydney, NSW 2109, Australia; andrew.care@mq.edu.au (A.C.); anwar.sunna@mq.edu.au (A.S.); andrei.zvyagin@mq.edu.au (A.Z.); 3Glytherix Ltd., Sydney, NSW 2113, Australia; douglas.campbell@mq.edu.au (D.C.); brad.walsh@minomic.com (B.W.); 4The Institute of Biology and Biomedicine, Lobachevsky State University of Nizhny Novgorod, 23 Gagarin ave., Nizhny Novgorod 603950, Russia; irin-b@mail.ru; 5The Institute of Molecular Medicine, I.M. Sechenov First Moscow State Medical University, 8-2 Trubetskaya str., Moscow 119991, Russia

**Keywords:** upconversion nanoparticles, photoluminescent nanocomplexes, bladder cancer, solid-binding peptide, glypican-1, photodynamic diagnostics

## Abstract

Bladder cancer is the ninth most common cancer worldwide. Due to a high risk of recurrence and progression of bladder cancer, every patient needs long-term surveillance, which includes regular cystoscopy, sometimes followed by a biopsy of suspicious lesions or resections of recurring tumours. This study addresses the development of novel biohybrid nanocomplexes representing upconversion nanoparticles (UCNP) coupled to antibodies for photoluminescent (PL) detection of bladder cancer cells. Carrying specific antibodies, these nanoconjugates selectively bind to urothelial carcinoma cells and make them visible by emitting visible PL upon excitation with deeply penetrating near-infrared light. UCNP were coated with a silica layer and linked to anti-Glypican-1 antibody MIL38 via silica-specific solid-binding peptide. Conjugates have been shown to specifically attach to urothelial carcinoma cells with high expression of Glypican-1. This result highlights the potential of produced conjugates and conjugation technology for further studies of their application in the tumour detection and fluorescence-guided resection.

## 1. Introduction

Bladder cancer is the fourth most common cancer in men, eleventh in women and one of the most expensive tumours to treat [1,2]. Despite the considerable progress in biotechnology and medicine over the past few decades, the recurrence and progression rates of bladder cancer remain significant and emphasise a need for novel methods of bladder cancer diagnosis and therapy. Approximately three-quarters of bladder cancer patients initially present with a non-muscle invasive tumour, which is usually resected [3,4]. Transurethral resection can be followed by adjuvant chemo- and immune-therapy. However, in up to 50% of patients, aggressive flat lesions remain and progress into muscle-invasive disease, potentially leading to cystectomy and a less favourable outcome. Due to a high risk of recurrence and progression of bladder cancer, every patient needs long-term surveillance, which includes regular cystoscopy, sometimes followed by a biopsy of suspicious lesions or resections of recurring tumours. This surveillance, as well as a number of other aspects, makes bladder cancer one of the most expensive tumours to treat on a per patient basis [5,6]. A number of known reasons can cause recurrence or progression of bladder cancer despite active therapy. In general, it is usually caused by the incomplete initial treatment. It can be incomplete resection, which may lead to the development of remaining tumours and tumour cells, or ineffective adjuvant therapy, which may lead to reimplantation of cancer cells [7].

Recent knowledge about genetic characteristics, molecular basis and metabolic pathways of tumours [8,9,10] allowed the establishment of a model of a highly effective contemporary medical approach that should: (1) be based on the individual molecular profile of the disease in each patient, (2) involve various mechanisms of action on cancer cells and (3) allow monitoring of the treatment progress. These tasks can be carried by theranostic agents suitable for simultaneous molecular diagnostic and specific targeted therapeutic action on tumour cells with monitoring of treatment response [11]. These theranostic agents should be able to allow early detection, guide a surgeon during resection and deliver targeted therapeutic agents to kill remaining tumours and disturbed cancer cells. Therefore, they must consist of a targeting part that can bind to bladder tumour, a part that can produce bright luminescence and a part that can cause death of cancer cells.

The unique photophysical properties of upconversion nanoparticles (UCNP) allow optical imaging at the centimetre-depth in biological tissue, which is demanded for a number of biomedical applications [12]. Their photophysical properties provide high contrast of the labelled structures against the background of strong scattering and autofluorescence of biological tissue [13]. Upconversion photoluminescence (PL) converts NIR radiation to visible light by using sequential photons and requires low laser power densities (1–10^3^ W cm^−2^) to generate higher-energy visible photons [14]. Moreover, UCNP can preferentially accumulate in tumour tissue due to the leaky capillary blood vessels in tumours in virtue of enhanced permeability and retention effect (EPR) [15,16]. Coating with silica could allow incorporation of imaging or targeting agents or the most effective individually chosen drugs [17,18].

To become a multifunctional agent, UCNP need a tumour-targeting unit. Proteins [19,20], folic acid [21] and antibodies [22] were previously used for diagnostic and therapeutic actions of UCNP. Antibodies are preferable as targeting agents, as they will allow adjustment of the affinity of nanoconjugates, depending on the antigen expression of target cells. A monoclonal antibody MIL-38 had shown high affinity to Glypican-1, which is expressed by urinary bladder cancer cells and has a potential to deliver nanoconjugates to urothelial carcinoma cells [23]. Glypican-1 is a growth factor receptor and, therefore, participates in the control of growth and division of cells [24]. In addition to urothelial carcinoma, the presence of this proteoglycan was previously reported in prostate [25], pancreatic [26], oesophageal [27], breast [28] and brain [29] cancers.

The choice of a method of the bioconjugation of nanoparticles with antibodies is crucial, as it can affect functioning of antibody. Conventional methods of bioconjugation involving carboxyl activating agents or amine-reactive crosslinking suffer from complexity and poor coordination control of functional biomolecules [30,31]. In this study we applied a recently developed self-assembling bioconjugation strategy based on a silica-specific solid-binding peptide (linker) that exhibited the high binding affinity towards silica [32]. This linker can be genetically fused to a protein of interest and the resulting recombinant fusion protein (linker protein) binds strongly to silica-containing materials. Using genetic engineering, the linker was incorporated into the N-terminus of truncated form of antibody-binding Protein G of *Streptococcus* strain G148 [32]. This bifunctional fusion protein, Linker-Protein G (LPG), has been shown to act as an anchorage point for antibodies at the surfaces of silica-coated nanoparticles [30,33]. Specific binding to crystallisable fragment (Fc) of an antibody prevents interference with antigen-binding sites (Fab) and happens within minutes and without the need for any chemical modification or physical treatment [30].

In this study we report of designed targeted photoluminescent nanoconjugates based on silica-coated UCNP and functionalised with anti-Glypican-1 antibodies. This novel photoluminescent nanoconjugates is suitable for targeted labelling of urothelial carcinoma cells and conjugation technology is promising to produce multifunctional agents for early detection, fluorescence-guided resection of non-muscle-invasive bladder cancer.

## 2. Materials and Methods

### 2.1. Synthesis of UCNP

UCNP were synthesised by solvatothermal decomposition method [34]. YCl_3_ (0.8 mmol), YbCl_3_ (0.18 mmol) and ErCl_3_ (0.02 mmol) were added to a flask containing 6 mL of oleic acid and 15 mL of octadecene, heated to 160 °C for 30 min to dissolve the lanthanide salts under an argon flow. After the heating, the mixture was cooled to room temperature. NaOH (2.5 mmol) and NH_4_F (4 mmol) dissolved in 10 mL of methanol were then added to the flask and stirred for 30 min at room temperature. Subsequently, this mixture was heated to 110 °C for 30 min to remove the residual methanol and water. For the following 1 h, this mixture was heated to 310 °C under argon flow with stirring and cooled to room temperature afterwards. UCNP were washed three times with ethanol/methanol (1:1, *v*/*v*) solution and suspended in cyclohexane to obtain the particle suspension.

Silica coating was performed as follows: 5 mL of NaYF_4_:Yb,Er (0.1 mmol) cyclohexane suspension and 5 mL of Igepal CO-520 (0.5 mL) cyclohexane solution were mixed in a one-neck flask and kept sealed and stirring at room temperature for 3 h. After that, 500 μL of ammonium hydroxide solution was added to the mixture and kept stirring for another 2 h. The following step was the slow injection of 40 μL of tetraethyl orthosilicate (2 μL/min) to the reaction mixture. The stirring was continued for 24 h before the addition of ethanol to precipitate the nanoparticles. The obtained silica-coated UCNP (UCNP@SiO_2_) were then washed three times with 100% ethanol and another three times with ultrapure water. 

### 2.2. Production of the UCNP Nanoconjugates

LPG linker was kindly provided by Andrew Care and Anwar Sunna (Department of Chemistry and Biomolecular science and ARC Centre of Excellence for Nanoscale BioPhotonics, Macquarie University). Before conjugation with LPG, silica-coated UCNP were washed three times with 100% ethanol and another three times with ultrapure water. Then, UCNP were washed three times with 100 mM Tris-HCl (Ph = 7.5) and resuspended in 400 µL of Tris buffer containing 20 μg of LPG. After that, this mixture was agitated at 4 °C for 30 min and LPG bound nanoconjugates (UCNP@SiO_2_-LPG) were collected by centrifugation at 7000× *g* at 4 °C. UCNP@SiO_2_-LPG were washed two times with Tris buffer at 4 °C.

Glypican-1 monoclonal antibody MIL-38 was produced and kindly provided by Minomic International Ltd. (Sydney, Australia). Cryptosporidium monoclonal antibody CRY104 was kindly provided by A. Sunna and A. Care. UCNP@SiO_2_-LPG nanoconjugates were incubated with antibody MIL-38 or CRY104 with ratio of 20 μg of the antibody per 1000 μg of UCNP with agitation for 30 min at 4 °C. Nanoconjugates UCNP@SiO_2_-LPG-MIL-38/CRY104 were separated from unbound antibodies by centrifugation and two washings with Tris buffer. Resulting nanoconjugates were redispersed in 1040 μL of Tris buffer to produce the 1 mg/mL suspension.

### 2.3. Characterisation of UCNP/UCNP@SiO_2_/Nanoconjugates

Transmission electron microscopy of UCNP/UCNP@SiO_2_ was performed using a CM10 electron microscope (Philips, Eindhoven, Netherlands). Size distribution of UCNP/UCNP@SiO_2_ was analysed by using the ImageJ software (1.47v, National Institute of Mental Health, Bethesda, MD, USA). Hydrodynamic diameter of UCNP@SiO_2_/nanoconjugates by dynamic light scattering and zeta potential [35] were measured on a Zetasizer Nano ZS90 (Malvern instruments Ltd., Malvern, UK). The intensity of the photoluminescence of UCNP was measured using a spectrofluorometer Fluorolog-Tau3 (HORIBA Jobin Yvon GmbH, Bensheim, Germany) equipped with an external 978-nm fibre-coupled diode laser (ATC-Semiconductor devices, St. Petersburg, Russia).

### 2.4. Cell Labelling

T24 and C3 urothelial carcinoma cell lines were kindly provided by Minomic International Ltd. Affinity of the monoclonal antibody MIL-38 towards urothelial carcinoma cells T24 was described previously [36] and was confirmed by the results of the flow cytometry analysis (Appendix A).

T24 and C3 cells were cultured in a RPMI 1640 Medium supplemented with 10% and 20% of FBS, respectively. In a 24-well plate, 2 × 10^5^ T24 and C3 cells per well were seeded on glass coverslips coated by Poly-D-lysine hydrobromide (Sigma-Aldrich, Saint Louis, MO, USA). After incubation for 24 h at 37 °C and 5% CO_2_, cells were washed three times by PBS and fixed by incubation for 20 min with 200-µL of 4% paraformaldehyde. Then, cells were washed with PBS three more times and refrigerated for 24 h before the targeted labelling experiment.

Four groups were formed: positive group: T24 + UCNP-LPG-MIL38; three negative controls: C3 + UCNP-LPG-MIL38 (negative cell line), T24 + UCNP-LPG-CRY104 (negative antibody), T24 + UCNP-LPG (no antibody). Nanoconjugates suspensions with concentrations of 25 µg/mL and 500 µL in Tris-buffered saline were added to the cells. After incubation for 1 h, coverslips were thoroughly washed with PBS and mounted of slides using ProLong Diamond Antifade Mountant with DAPI (Thermo Scientific). The imaging of labelled cells was performed by using a confocal laser-scanning microscope Zeiss LSM880 (Carl Zeiss AG, Oberkochen, Germany) with oil-immersion objective with 60× magnification and numerical aperture NA 1.4., equipped with 405-nm and 980-nm lasers. For quantification of labelled cells and analysis of photoluminescence intensity, one coverslip was taken from each group, and regions from 1.37 to 3.19 mm^2^ were scanned in order to detect at least 110 cells in each group. ZEN imaging software was used to analyse photoluminescence characteristics of labelled cells. Percentages of the labelled cells and photoluminescence intensity of T24 and C3 labelled cells were compared between the groups. 

## 3. Results

### 3.1. Production and Characterisation of Targeted Nanoconjugates UCNP@SiO2-LPG-MIL38

UCNP of the composition NaYF4:Yb:Er were chosen as a platform for this research. They were produced and coated with silica to become amenable for further conjugation with biomolecules. To target T24 urothelial carcinoma cells, expressing Glypican-1, silica-coated UCNP (UCNP@SiO_2_) were conjugated with a Glypican-1 monoclonal antibody MIL-38, by using LPG linker, resulting in targeted upconversion nanoconjugates UCNP@SiO_2_-LPG-MIL-38 (Figure 1). 

UCNP of the composition NaYF_4_, doped with 18% Yb and 2% Er (NaYF_4_:Yb,Er), were synthesised by solvothermal decomposition method [34]. UCNP had spherical shape, were monodispersed (Figure 2A) and had a mean size of 33.1 nm with a narrow size distribution (standard deviation = 1.5 nm) (Figure 2C).

UCNP were then coated with a layer of silica using water-in-oil microemulsion method [37]. Silica coating was chosen because of its ability to convert the hydrophobic nanoparticles into hydrophilic and improve their colloidal stability in water and in physiological buffers. Additionally, the silica layer allows efficient functionalisation of these nanoparticles with other molecules, such as photosensitisers for photodynamic therapy (PDT), antibodies for targeted action and immunomodulation or various drugs to use UCNP as nanocarriers [17,38,39].

Transmission electron microscopy demonstrated the successful coating with silica, resulting in retaining of dispersion of UCNP (Figure 2B). An increase of their average diameter from 33.1 ± 1.5 nm to 41.2 ± 2.4 nm (Figure 2D) demonstrated that the thickness of the silica layer was approximately 4 nm. The shape of the UCNP after silica coating remained spherical (Figure 2A,B). Resulting UCNP s displaced a negative zeta-potential of –16.6 mV (Table 1) and mean hydrodynamic diameter of 227 nm, which was probably affected by a limited number of large clusters, whereas the vast majority of UCNP had a hydrodynamic diameter around 100 nm (Figure 2E).

One of the most advantageous properties of the silica-coated UCNP is upconversion PL. These nanoparticles emit bright green light under the illumination with a 980-nm near-infrared laser. The upconversion PL of UCNP featured two bands in green (535–555 nm) and red (645–679 nm) regions of the emission spectrum (Figure 2F). Peaks of green light at 550 nm and red light at 650 nm are characteristic to Yb/Er co-doped UCNP [40]. The optical properties of UCNPs were largely preserved after silica coating, and particles displayed satisfactory brightness for the subsequent observation of cellular labelling.

In order to conjugate UCNP with antibodies, nanoparticles firstly had to be functionalised with LPG linker [30,32,41]. The affinity of the solid-binding peptides towards solid surfaces is caused by a combination of hydrophobic, electrostatic, polar and other multiple non-covalent interactions [30,32]. The negative charge of the silica surfaces and positive charge of the used linker with a sequence of (VKTQATSREEPPRLPSKHRPG)_4_VKTQTAS made by basic lysine and arginine residues underpins this binding [41]. Moreover, an intrinsic structural disorder of this linker increases its flexibility and plasticity. As a result, it further promotes electrostatic interactions between positively charged residues of the peptide and negatively charged silica-containing nanomaterials [32,41]. Protein G was genetically fused with a solid-binding peptide linker, resulting in a Linker-Protein G (LPG) with a region binding to silica-containing materials, such as silica-coated UCNP, and a region that allows oriented immobilisation of Ig-G antibodies, such as anti-Glypican-1 monoclonal Ig-G antibody MIL-38. Functionalisation by a positively charged LPG was confirmed by a reduction of the negative zeta potential charge of the nanoparticles from −16.6 mV (UCNP@SiO_2_) to −9.65 mV (UCNP@SiO_2_-LPG) (Table 1).

MIL-38 is a monoclonal antibody that was initially raised against urothelial carcinoma cells and was previously known as BLCA-38 [23]. In this study, the specificity of anti-Glypican-1 antibody MIL-38 towards urothelial carcinoma cells was utilised for their targeted labelling by UCNP coupled to MIL-38 and termed targeted UCNP nanoconjugates. The anchoring of LPG on the surface of UCNP allowed their simple bioconjugation with monoclonal antibodies MIL-38. The success of the conjugation was confirmed by a positive shift of the zeta potential of UCNP@SiO_2_-LPG-MIL-38 to −5.75 mV (Table 1) and an increase of the hydrodynamic diameter of nanoconjugates (Figure 2E). The decrease of the surface charge, following the binding of LPG, was noted, which can be explained by the highly positive charge of LPG in neutral pH (isoelectric point of LPG, 11.2), which compensated the negative charge of the silica-coated UCNP. The binding of MIL-38 further decreased the absolute value of the surface charge, which can be explained by the thickening of Stern layer due to the antibody attachment [42]. Upconversion PL intensity decreased slightly after the bioconjugation with antibody (Figure 2F). However, the emission spectrum remained unchanged, with targeted upconversion nanoconjugates UCNP@SiO_2_-LPG-MIL38 featuring the characteristic green and red PL bands under the 980-nm excitation.

### 3.2. Interaction of UCNP Nanoconjugates with Urothelial Carcinoma Cells

Assessment of sensitivity and specificity of the binding of nanoconjugates UCNP@SiO_2_-LPG-MIL-38, was performed by incubation with urothelial carcinoma cells with high expression of Glypican-1 (T24) and urothelial carcinoma cells with low expression of Glypican-1 (C3). To find out the role of the antibody MIL-38, other controls consisted of incubation of T24 cells with nanoconjugates UCNP@SiO_2_-LPG-CRY104, functionalised by control isotype cryptosporidium antibody CRY104 and with nanoconjugates UCNP@SiO_2_-LPG, without an antibody. Confocal laser-scanning microscopy was used to detect a representative number of cells in each group (Figure 3).

The microscopy imaging demonstrated binding of the targeted nanoconjugates functionalised by a Glypican-1 monoclonal antibody MIL-38 to the vast majority of T24 cells. Incubation of T24 cells with UCNP@SiO_2_-LPG-CRY-104 and UCNP@SiO_2_-LPG resulted in only few cells being labelled. Nanoconjugates were observable in all groups, although they were predominantly found adhered to the plate (perceived as background) rather than on the cells in the control groups. Visual quantification was then performed to separate cells within each treatment group into those labelled with nanoconjugates and unlabelled and to calculate their proportions. Finally, the percentages of the labelled cells were compared between the groups (Figure 4A).

Visual quantification showed that 90% of T24 cells incubated with nanoconjugates UCNP@SiO_2_-LPG-MIL-38 were found labelled. The incubation of the same nanoconjugates with the control C3 cells resulted in labelling of only 23.2%, which was probably caused by the low expression of Glypican-1. Among the T24 cells incubated with nanoconjugates with the control antibodies (UCNP@SiO_2_-LPG-CRY104) and incubated with nanoconjugates without antibodies (UCNP@SiO_2_-LPG), only 19.8% and 26.2% cells were labelled.

These results proved the specificity of the binding of the targeted nanoconjugates UCNP@SiO_2_-LPG-MIL-38, which predominantly bound to Glypican-1 positive T24 cells. This result shows that MIL-38 antibody has a high potential in targeted delivery of biohybrid nanocompounds to T24 urothelial carcinoma cells expressing Glypican-1.

The labelling performance was also characterised by measuring the PL intensity of cells in different groups after incubation with photoluminescent nanoconjugates (Figure 4B). The mean PL intensity of all T24 cells incubated with nanoconjugates UCNP@SiO_2_-LPG-MIL-38 was found to be almost eight times higher than the mean PL intensity of C3 cells incubated with the same nanoconjugates and was more than five times higher than mean PL intensity of T24 cells incubated with control nanoconjugates UCNP@SiO_2_-LPG-CRY104 or UCNP@SiO_2_-LPG. Such a significant difference in the PL intensity of cells from the tested groups further demonstrated considerable potential of photoluminescent nanoconjugates UCNP@SiO_2_-LPG-MIL-38 for targeting and binding to T24 urothelial carcinoma cells.

## 4. Discussion

Photodynamic diagnostics and PDT of bladder cancer show considerable promise in bladder cancer treatment, as they can act at the causes of its recurrence and progression. However, the existing photosensitisers are associated with a number of adverse side effects, have low specificity and their treatment depth is limited to superficial layers of tumours. They also suffer from inability of proper targeting and adjustment of therapeutic effect. Biohybrid photoluminescent nanoconjugates based on the UCNP are promising for solving the problem of the greater treatment depth by using near-infrared light to visualize cancer cells and photoactivate photosensitisers [43]. This can lead to the reduction of recurrence and progression rates. Moreover, UCNP nanoconjugates are capable to carry different targeting modules and therapeutic agents (e.g., chemo- and radiotherapeutic agents, protein toxins) to combine modalities of action on cancer cells optimised for certain type of tumour.

In this study, UCNP were conjugated with anti-Glypican-1 monoclonal antibodies MIL-38 to target urothelial carcinoma cells T24. Obtained nanoconjugates are intended for tumour targeting including passive targeting by EPR-effect and active targeting by antibodies. The main advantage of passive targeting is that injected intravenously nanoparticles of size 400 nm and under can accumulate in a tumour without any specific targeting agents. In tumours of the urinary bladder, passive accumulation of nanoparticles in larger tumours may be possible using systemic intravenous injection. Active targeting of nanoparticles is usually based on an antibody, antibody fragments, aptamers or ligand-based targeting [44]. Binding to an antibody was the first and widespread approach of targeting nanoparticles [45]. In comparison with the other targeting agents, antibodies do not require modification and are amenable for easy conjugation using a number of protocols. One of the noteworthy protocols based on LPG linker ensures facile and universal attachment of antibody to all types of nanoparticles surface-coated with a silica layer. The presence of fragments with different functions in an antibody allows oriented coupling to a nanoparticle, where Fc fragment binds to a nanoparticle and Fab fragments remain available for binding to a specific antigen. As such, the oriented binding provides high efficacy and low risk of aggregation, which is oftentimes caused by binding of a single antibody molecule to several nanoparticles.

We produced UCNP amenable for conjugation with biomolecules. These nanoparticles had an average diameter of 33.1 ± 1.5 nm and exhibited upconversion PL. We used the silica-coating to apply LPG-mediated conjugation of UCNP with MIL-38, following a previously published protocol, which was developed by Liang et al. [40]. This protocol was found to be highly efficient and non-laborious, as the loss of nanoparticles and antibodies was minimal, and conjugated antibodies had their Fab fragments available for binding to their antigens. The increase of the mean hydrodynamic diameter of the antibody-coupled targeted upconversion nanoconjugates UCNP@SiO_2_-LPG-MIL38 can be interpreted by the attachment of MIL-38 to the surface of nanoparticles and by some degree of aggregation (as can be seen by a secondary peak centred at ~700 nm—see Figure 2E). This aggregation was likely attributed to the slight decrease of the overall surface charge, which provided insufficient electrostatic repulsion to prevent the close-range nanoparticle interaction.

The incubation of urothelial carcinoma cells T24 with targeted nanoconjugates UCNP@SiO_2_-LPG-MIL-38 resulted in labelling of almost 90% of cells and PL of targeted cells with the intensity up to eight times higher than in the control groups. Moreover, a series of control experiments demonstrated that specificity of this photoluminescent labelling was provided by MIL-38 monoclonal antibody. These results demonstrating the high sensitivity and specificity of nanoconjugates UCNP@SiO_2_-LPG-MIL-38 suggests that these nanoconjugates have a great potential in fluorescence cystoscopy, fluorescence-guided resection and PDT of bladder cancer.

## 5. Conclusions

In this study, we demonstrated the production of targeted upconversion photoluminescent nanoconjugates UCNP@SiO_2_-LPG-MIL-38 for photodynamic diagnosis of bladder cancer cells and assessed their selectivity and molecular specificity towards Glypican-1 positive urothelial carcinoma cells T24. These nanoconjugates specifically labelled targeted cells and made them observable upon the excitation with a near infrared laser. It was found that the monoclonal antibody MIL-38 holds promise for diagnosis, drug delivery and targeted therapy, as it mediated the targeted binding of upconversion photoluminescent nanoconjugates to Glypican-1 positive urothelial carcinoma cells. These are significant results, showing a high potential of these nanoconjugates for further exploration of photodynamic diagnosis.

## Figures and Tables

**Figure 1 biomolecules-09-00820-f001:**
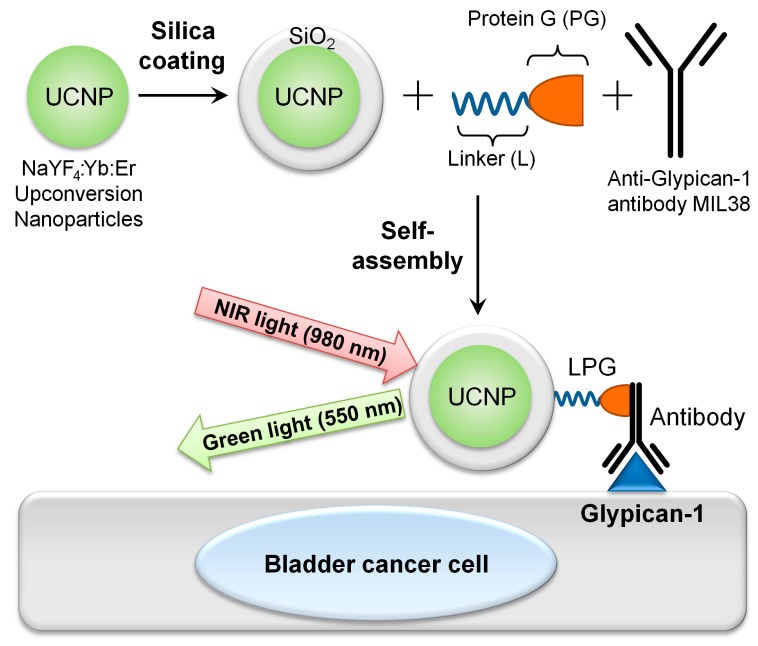
Schematic illustration of production and application of targeted upconversion nanoconjugates UCNP@SiO_2_-LPG-MIL-38 in labelling of bladder cancer cells.

**Figure 2 biomolecules-09-00820-f002:**
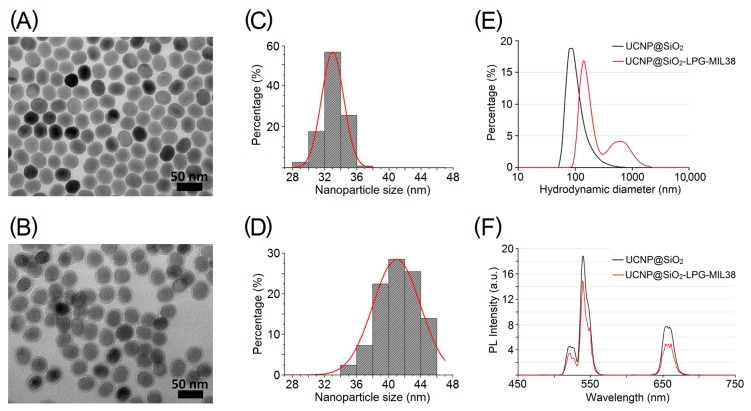
Properties of upconversion nanoparticles (UCNP) and nanoconjugates. (**A**–**B**) TEM images of UCNP (NaYF_4_:Yb,Er) before (**A**) and after (**B**) the coating with a silica layer; (**C**–**D**) size distribution of UCNP before (**C**) and after (**D**) the silica coating; (**E**) hydrodynamic diameter of silica-coated UCNP (UCNP@SiO_2_) and targeted nanoconjugates (UCNP@SiO_2_-LPG-MIL38); (**F**)PL emission spectra of UCNP@SiO_2_ and UCNP@SiO_2_-LPG-MIL38 under 980-nm excitation.

**Figure 3 biomolecules-09-00820-f003:**
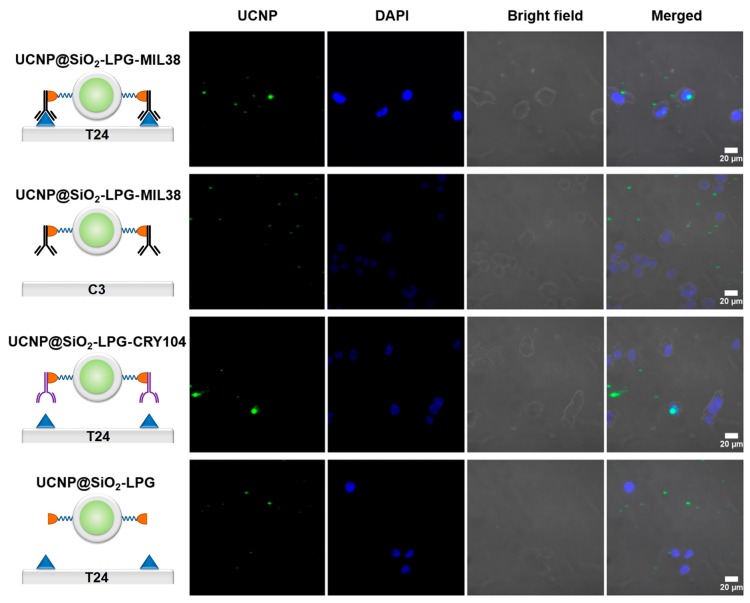
Interaction of UCNP nanoconjugates with urothelial carcinoma cells. Microscopy laser-scanning confocal PL images of Glypican-1 high T24 and Glypican-1 low C3 cells incubated with targeted and non-targeted nanoconjugates, magnification 60×. Columns: UCNP—PL signal in the range of 495–634 nm, excitation on 980 nm; DAPI—fluorescent signal in the range of 410–495 nm, excitation on 405 nm; bright-field microscopy images; merge of these three channels.

**Figure 4 biomolecules-09-00820-f004:**
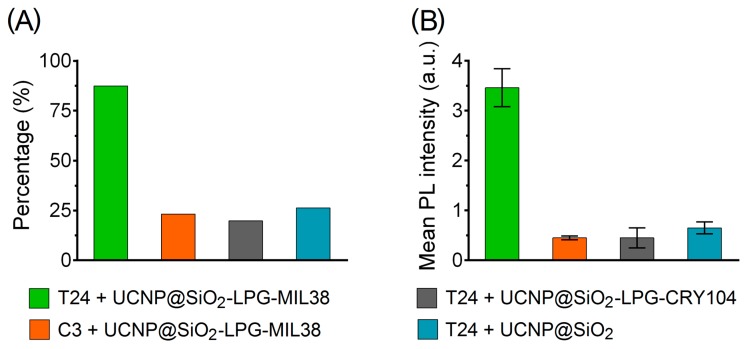
Interaction of UCNP nanoconjugates with urothelial carcinoma cells. (**A**) Percentage of Glypican-1 high T24 and Glypican-1 low C3 urothelial carcinoma cells labelled by targeted and non-targeted nanoconjugates; (**B**) mean PL intensity of T24 and C3 cells labelled by targeted and non-targeted nanoconjugates. Error bars represent the 95% confidence interval of the mean.

**Table 1 biomolecules-09-00820-t001:** Zeta-potentials of silica-coated UCNP and nanoconjugates.

UCNP/Nanoconjugate	Zeta Potential, mV
UCNP@SiO_2_	−16.6
UCNP@SiO_2_-LPG	−9.65
UCNP@SiO_2_-LPG-MIL-38	−5.75

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
