# Peer review of "Functionalized Upconversion Nanoparticles for Targeted Labelling of Bladder Cancer Cells"

_biomolecules, 2019, doi:10.3390/biom9120820_

Round 1
Reviewer 1 Report
The article “Functionalized upconversion nanoparticles for targeted labeling of bladder cancer cells” by D. Polikarpov et al. is devoted to the development of novel biohybrid nanocomplexes for photoluminescent (PL) detection of bladder cancer cells. The article is well written (although language editing is required) and easy to follow. It could be of interest to the broad scientific audience. The article can be published if the following shortcomings were addressed:
Major shortcomings:
Emission light (550nm) is close to the hemoglobin absorption peak and highly absorbed in the tissue. The penetration depth in vascularized tissues is less than 1mm. So, in vivo applications of such particles are highly problematic. The second emission peak at 660nm has better penetration and is more promising. Such as authors target theranostic (in vivo) applications, they should address this issue.Minor shortcomings:
Lines 34-35: Statements like “Approximately three-quarters of bladder cancer patients initially present with a non-muscle invasive tumor, which is usually resected.” require references Lines 70-72 “A monoclonal antibody MIL-38 had shown high affinity to Glypican-1, which is expressed by urinary bladder cancer cells and has a potential to deliver nanoconjugates to urothelial carcinoma cells.” Requires references Line 98: Acronym “OA” must be defined on their first occurrence Lines 116-118: The sentence “LPG linker, which was kindly provided by Andrew Care and Anwar Sunna (Department of Chemistry and Biomolecular science and ARC Centre of Excellence for Nanoscale BioPhotonics, Macquarie University).” Is not complete. Line 121, 127: What does “this mixture was rotated at 4°C for 30 min” mean? Stirred? Please elaborate Line 136: “zeta potential” is not universally known metric and requires references Line 143: “Fig. A1”- Unusual figures numbering. Moreover, Fig 1 has just one panel Line 182-183: Not clear what does “resulting in improved dispersion” mean. Decreased? Lines 222-224: Statements like “The binding of MIL-38 further decreased the absolute value of the surface charge, which can be explained by the thickening of Stern layer due to the antibody attachment.” require references Lines 222-231: Result interpretations belong to the Discussion sectionAuthor Response
Dear Reviewer,
We would like to express our deep gratitude for careful evaluation of our manuscript and your competent comments. We have thoroughly revised the manuscript according to your remarks. Please find below the detailed description of the revisions (Reviewer’s remarks are presented in bold, our answers follow in plain text).
The article “Functionalized upconversion nanoparticles for targeted labeling of bladder cancer cells” by D. Polikarpov et al. is devoted to the development of novel biohybrid nanocomplexes for photoluminescent (PL) detection of bladder cancer cells. The article is well written (although language editing is required) and easy to follow. It could be of interest to the broad scientific audience. The article can be published if the following shortcomings were addressed:
Major shortcomings:
Emission light (550nm) is close to the hemoglobin absorption peak and highly absorbed in the tissue. The penetration depth in vascularized tissues is less than 1mm. So, in vivo applications of such particles are highly problematic. The second emission peak at 660nm has better penetration and is more promising. Such as authors target theranostic (in vivo) applications, they should address this issue.
Indeed, UCNP doped with Yb and Er ions are characterized by photoluminescence in the visible region of the spectrum, which does not fall into the transparency window of the biological tissue. At the same time, their bright photoluminescence in the region of 550 and 660 nm provides an effective study of the specificity of tumor cells imaging using laser scanning confocal microscopy. The used UCNP synthesis technology allows to vary the composition and ratio of doping components to achieve the desired photophysical characteristics. In particular, the additional UCNP doping with Tm ions makes it possible to obtain nanocomplexes with photoluminescence maxima both in visible and near-IR region (at about 800 nm), which allows to efficiently visualize tumor foci in in vivo studies [Guryev et al., PNAS, 2018; Guryev et al., Toxicological Sciences, 2019]. The tuning of the optical properties of UCNP by varying their composition will be applied in further animal studies.
Guryev, E.L.; Shilyagina, N.Y.; Kostyuk, A.B.; Sencha, L.M.; Balalaeva, I.V., Vodeneev, V.A., Kutova, O.M.; Lyubeshkin, A.V.; Yakubovskaya, R.I.; Pankratov, A.A.; Ingel, F.I.; Novik, T.S.; Deyev, S.M.; Ermilov, S.A.; Zvyagin, A.V. Preclinical Study of Biofunctional Polymer-Coated Upconversion Nanoparticles Toxicological Sciences, 2019, 170, 123-132.
Guryev, E.L.; Volodina, N.O.; Shilyagina, N.Y.; Gudkov, S.V.; Balalaeva, I.V.; Volovetskiy, A.B.; Lyubeshkin, A.V.; Sen', A.V.; Ermilov, S.A.; Vodeneev, V.A.; Petrov, R.V.; Zvyagin, A.V.; Alferov, Z.I.; Deyev, S.M. Radioactive (90Y) upconversion nanoparticles conjugated with recombinant targeted toxin for synergistic nanotheranostics of cancer. PNAS 2018, 115, 9690-9695.
Minor shortcomings:
Lines 34-35: Statements like “Approximately three-quarters of bladder cancer patients initially present with a non-muscle invasive tumor, which is usually resected.” require references
We added additional relevant references to main text and References section:
Line 36:
Approximately three-quarters of bladder cancer patients initially present with a non-muscle invasive tumour, which is usually resected [3,4].
Line 343:
Babjuk, M.; Burger, M.; Zigeuner, R.; Shariat S.F.; van Rhijn B.W.; Compérat, E, Sylvester, R.J.; Kaasinen, E; Böhle A.; Palou Redorta, J.; Rouprêt, M. EAU guidelines on non-muscle-invasive urothelial carcinoma of the bladder: update 2013. European urology 2013, 64, 639-653. Brausi, M.; Witjes, J.A.; Lamm, D.; Persad, R.; Palou, J.; Colombel, M.; Buckley, R.; Soloway, M.; Akaza, H.; Böhle, A. A review of current guidelines and best practice recommendations for the management of nonmuscle invasive bladder cancer by the International Bladder Cancer Group. J Urol, 2011, 186, 2158-2167.
Lines 70-72 “A monoclonal antibody MIL-38 had shown high affinity to Glypican-1, which is expressed by urinary bladder cancer cells and has a potential to deliver nanoconjugates to urothelial carcinoma cells.” Requires references
We added additional relevant reference to main text and References section:
Line 72:
A monoclonal antibody MIL-38 had shown high affinity to Glypican-1, which is expressed by urinary bladder cancer cells and has a potential to deliver nanoconjugates to urothelial carcinoma cells [23].
Line 399:
Walker, K. Z.; Russell, P. J.; Kingsley, E. A.; Philips, J.; Raghavan, D. Detection of Malignant-Cells in Voided Urine from Patients with Bladder-Cancer, a Novel Monoclonal Assay. J Urology, 1989142, 1578-1583.
Line 98: Acronym “OA” must be defined on their first occurrence
Corrected
Line 98:
UCNP were synthesised by solvatothermal decomposition method [31]. YCl3 (0.8 mmol), YbCl3 (0.18mmol) and ErCl3 (0.02mmol) were added to a flask containing 6 mL of oleic acid and 15 mL of octadecene, heated to 160 °C for 30 min to dissolve the lanthanide salts under an argon flow.
Lines 116-118: The sentence “LPG linker, which was kindly provided by Andrew Care and Anwar Sunna (Department of Chemistry and Biomolecular science and ARC Centre of Excellence for Nanoscale BioPhotonics, Macquarie University).” Is not complete.
Corrected
Line 116:
LPG linker was kindly provided by Andrew Care and Anwar Sunna (Department of Chemistry and Biomolecular science and ARC Centre of Excellence for Nanoscale BioPhotonics, Macquarie University).
Line 121, 127: What does “this mixture was rotated at 4°C for 30 min” mean? Stirred? Please elaborate
Corrected
Line 121:
After that, this mixture was agitated at 4°C for 30 min and LPG bound nanoconjugates (UCNP@SiO2-LPG) were collected by centrifugation at 7000 × g at 4 ºC.
Line 127:
UCNP@SiO2-LPG nanoconjugates were incubated with antibody MIL-38 or CRY104 with ratio of 20 μg of the antibody per 1000 μg of UCNP with agitation for 30 min at 4°C.
Line 136: “zeta potential” is not universally known metric and requires references
We added additional relevant reference to main text and References section:
Line 135:
Hydrodynamic diameter of UCNP@SiO2/nanoconjugates by dynamic light scattering and zeta potential [35] were measured on a Zetasizer Nano ZS90 (Malvern instruments Ltd.).
Line 427:
Bhattacharjee, S. DLS and zeta potential - What they are and what they are not? J Control Release, 2016, 235:337-351.
Line 143: “Fig. A1”- Unusual figures numbering. Moreover, Fig 1 has just one panel
The “Fig.A1” link meant the Fig. in the Appendix section, not in the main text.
To avoid confusion, the text and the figure from the Appendix section were moved to Supplementary Materials section.
The corresponding correction made in the text.
Line 142:
Affinity of the monoclonal antibody MIL-38 towards urothelial carcinoma cells T24 was described previously [36] and was confirmed by the results of the flow cytometry analysis (Fig. S1).
Line 337:
Supplementary Materials: The following are available online at: …, Figure S1. Flow cytometry analysis demonstrated the binding of MIL-38 antibody to T24 cells and minimal binding to C3 cells.
Line 182-183: Not clear what does “resulting in improved dispersion” mean. Decreased?
The corresponding correction made in the text.
Line 182:
Transmission electron microscopy demonstrated the successful coating with silica, resulting in retaining of dispersion of UCNP (Fig. 2B).
Lines 222-224: Statements like “The binding of MIL-38 further decreased the absolute value of the surface charge, which can be explained by the thickening of Stern layer due to the antibody attachment.” require references
We added additional relevant reference to main text and References section:
Line 222:
The binding of MIL-38 further decreased the absolute value of the surface charge, which can be explained by the thickening of Stern layer due to the antibody attachment [43].
Line 447:
Jacobs, M.; Selvam, A.P.; Craven, J.E.; Prasad, S. Antibody-Conjugated Gold Nanoparticle Based Immunosensor for Ultra-Sensitive Detection of Troponin-T. Journal of Laboratory Automation, 2014, 19, 546–554.
Lines 222-231: Result interpretations belong to the Discussion section
Indicated sentences have been moved to the Discussion section.
Line 306:
The increase of the mean hydrodynamic diameter of the antibody-coupled targeted upconversion nanoconjugates UCNP@SiO2-LPG-MIL38 can be interpreted by the attachment of MIL-38 to the surface of nanoparticles and by some degree of aggregation (as can be seen by a secondary peak centred at ~700 nm - see Fig. 2E). This aggregation was likely attributed to the slight decrease of the overall surface charge, which provided insufficient electrostatic repulsion to prevent the close-range nanoparticle interaction.
Reviewer 2 Report
The manuscript by Polikarpov et al. reports an antibody-conjugated upconversion NPs for labeling of bladder cancer cells. The topic is important, the experiments were properly designed, and the results are generally clear. The manuscript is well-written as well. There're a few comments for the authors to consider.
It's highly recommended that the in vivo animal tumor targeting study to be carried out. Many NPs failed to work on animals although their cellular results are good. The dynamic light scattering data showed a population of large aggregates of NPs. In those NPs, the surface area is small and may have strong non-specific interactions with cells. It's recommended that this population to be removed (by centrifugation or other techniques). What were the estimated number of antibodies on NP surface?Author Response
Dear Reviewer,
We would like to express our sincere appreciation for your careful attention to our manuscript and for the suggested improvements and valuable comments. Please find below the detailed answers to the questions mentioned in your review (Reviewer’s remarks are presented in bold, our answers follow in plain text).
The manuscript by Polikarpov et al. reports an antibody-conjugated upconversion NPs for labeling of bladder cancer cells. The topic is important, the experiments were properly designed, and the results are generally clear. The manuscript is well-written as well. There're a few comments for the authors to consider.
It's highly recommended that the in vivo animal tumor targeting study to be carried out. Many NPs failed to work on animals although their cellular results are good. The dynamic light scattering data showed a population of large aggregates of NPs. In those NPs, the surface area is small and may have strong non-specific interactions with cells. It's recommended that this population to be removed (by centrifugation or other techniques). What were the estimated number of antibodies on NP surface?
We agree with the reviewer of the need for in vivo studies for the unambiguous conclusion concerning the applicability of established nanocomplexes for fluorescent diagnosis of bladder cancer. This paper presents a universal technology of self-assembly of the teranostic nanocomplexes and the results of the first stage of the study of the potential for their application. In the future, we plan to carry out the necessary studies on tumor-bearing animals that will prove the effectiveness of the developed nanocomplexes.
In the presented study, we did not remove the formed particle aggregates to assess the colloidal stability of the created nanocomplexes and to analyze the effect of aggregation on the specificity of the attachment of nanocomplexes to target cells. As can be seen from the results, the specificity of labeling of urothelial carcinoma cells remained at a high level. In future studies on tumor-bearing animals the possible aggregates of nanocomplexes will be removed from the suspension by a well-established technique using low speed centrifugation. The concentration of nanocomplexes that retained colloidal stability will be determined by the intensity of the UCNP photoluminescence allowing precise control of the dose administered to animals.
According to our data, ~96% of antibody molecules were attached to UCNP and this corresponds to ~13 antibody molecules per particle. Previously, both we and other researchers have shown that this ratio of antibodies to nanoparticles is sufficient for efficient visualization of target cells [Wang et al., ACS Nano 2009; Liang et al., Acs Appl Mater Inter 2016].
Liang L.; Care A.; Qian Y.; Zhang R.; Packer N.H.; Sunna A.; Lu Y.; Zvyagin A.V. Facile Assembly of Functional Upconversion Nanoparticles for Targeted Cancer Imaging and Photodynamic Therapy. Acs Appl Mater Inter 2016, 8, 11945-11953.
Wang M., Mi C., Wang W., Liu C., Wu Y., Xu Z., Mao C., Xu S. Immunolabeling and NIR-Excited Fluorescent Imaging of HeLa Cells by Using NaYF4:Yb,Er Upconversion Nanoparticles / ACS Nano 2009, 23;3(6):1580-6.
Reviewer 3 Report
The description of characterization of nanopartiles and nanocomplex should be better presented. The UV-Vis of obtained nanoparticles and nanocomples should be performed. The FTIR spectra of nanoparticles and nanocomplex should be perform to obtain information about places of connection the antibodies with nanoparticles.Author Response
We would like to express our deep gratitude for sincere appreciation for your careful attention to our manuscript. Please find below the detailed answers to the questions mentioned in your review (Reviewer’s remarks are presented in bold, our answers follow in plain text).
The description of characterization of nanopartiles and nanocomplex should be better presented. The UV-Vis of obtained nanoparticles and nanocomples should be performed. The FTIR spectra of nanoparticles and nanocomplex should be perform to obtain information about places of connection the antibodies with nanoparticles.
The UV-Vis method is mainly used to determine low molecular weight compounds loaded or attached to UCNP [Dai et al., Biomaterials, 2012; Sun et al., Chemical Engineering & Processing: Process Intensification, 2019]. Nanoparticles are generally very reflective. It is difficult to use UV-Vis to characterize the difference between nanoparticles and nanocomplexes and confirm the presence of antibody. Therefore, we did not use this method to characterize the nanoparticles and nanocomplexes.
FTIR spectroscopy is informative for analysis of low molecular weight ligands and indeed allows to reveal the groups participating in the ligand-to-NP interaction. In the case of proteins, the size and complexity of the molecule do not allow such a detailed analysis. FTIR of proteins provides information on the secondary molecular folding structure. This method can confirm the fact of the protein-to-NP interaction, however, it seems unlikely to distinguish characteristic absorption bands due to the participation of certain groups of the protein molecule. Also, it is not possible to reliably justify changes in the FTIR spectrum if they are detected. Therefore, this method was not used in the analysis.
Dai Y. L., Yang D. M., Ma P. A., Kang X. J., Zhang X., Li C. X., Hou Z. Y., Cheng Z. Y., Lin J., Doxorubicin conjugated NaYF4:Yb3+/Tm3+ nanoparticles for therapy and sensing of drug delivery by luminescence resonance energy transfer / Biomaterials, 2012, 33, 8704–8713.
Sun X., Zhang P., Hou Y., Li Y., Huang X., Wang Z., L Jing L., Gao M. Upconversion luminescence mediated photodynamic therapy through hydrophilically engineered porphyrin / Chemical Engineering & Processing: Process Intensification, 2019, 142, 107551.